# In Vivo Imaging of Rabbit Follicles through Combining Ultrasound Bio-Microscopy and Intravital Window

**DOI:** 10.3390/ani14121727

**Published:** 2024-06-07

**Authors:** Lihan Yang, Chang Yan, Siming Tao, Yifeilong He, Jing Zhao, Yanya Wang, Yingjie Wu, Ning Liu, Yinghe Qin

**Affiliations:** 1State Key Laboratory of Animal Nutrition and Feeding, China Agricultural University, Beijing 100193, China; s20213040658@cau.edu.cn (L.Y.); 19825531012@163.com (C.Y.); 18045159167@163.com (S.T.); heyifeilong@163.com (Y.H.); z1678057869@163.com (J.Z.); 15187385580@163.com (Y.W.); wuyingjie@cau.edu.cn (Y.W.); 2National Engineering Laboratory for Animal Breeding, China Agricultural University, Beijing 100193, China

**Keywords:** ultrasound bio-microscopy, intravital window, follicle, female rabbit

## Abstract

**Simple Summary:**

Ovarian translocation and intravital window implantation surgery contribute to high-resolution imaging of the ovarian structure of female rabbits in vivo using UBM. High-resolution UBM could effectively detect the ovarian structure and measure the size of follicles. The combination of UBM and intravital window implantation provides a chance to achieve real-time dynamic and continuous imaging of the ovary. This study offers a relatively novel method to visualize ovarian dynamics, ovulation induction, and the cumulus–oocyte complex in vivo instead of conventional histological examination.

**Abstract:**

Continuous ovarian imaging has been proven to be a method for monitoring the development of follicles in vivo. The aim of this study was to evaluate the efficacy of combining ultrasound bio-microscopy (UBM) with an intravital window for follicle imaging in rabbits and to monitor the ovarian dynamic processes. New Zealand White female rabbits (n = 10) received ovarian translocation to a subcutaneous position. The ovarian tissue was sutured onto the abdominal muscles and covered with an intravital window for the continuous monitoring of the follicles using UBM. Results show that physiological changes (red blood cell and white blood cell counts, feed intake, and body weight change) in rabbits induced by surgery returned to normal physiological levels in one week. Furthermore, UBM could provide high-resolution imaging of follicles through the intravital window. Daily monitoring of ovarian dynamic processes for 6 days displayed variabilities in follicle counts and size. Collectively, these results provide a relatively new method to monitor ovarian dynamic processes and to understand the reproductive physiology of female rabbits.

## 1. Introduction

Rabbits are three-purpose animals widely used as farm animals (for meat and fur production) [1], experimental animals [2], and pets [3]. Rabbits are stimulated ovulating animals, which are good animal models for investigating the underlying mechanism of inducing ovulation and follicle development [4]. Generally, follicle development in rabbits is documented by histology [5,6], laparotomy [7,8], laparoscopic inspections [9], and endocrinology [10,11]. However, these methods seldom provide opportunities for continuous examination, which limits a deep understanding of rabbit follicular development and ovulation. Therefore, an in vivo continuous imaging method is needed for ovarian function research in female rabbits. B-mode ultrasonography is widely used to examine the dynamics of ovarian follicles and corpus luteum in cattle [12], camels [13], sheep [14,15], and humans [16,17]. In mice and rabbits, B-mode ultrasound is preferred for performing pregnancy diagnoses and antenatal examinations [18]. However, the failure to use B-mode ultrasonography in monitoring small anatomical structures is attributed to the low resolution of ultrasounds with 5.0~7.5 MHz [19]. 

Ultrasound bio-microscopy (UBM), a high-resolution ultrasound technique, utilizes 50~100 MHz ultra-high-frequency sound waves to produce a very high-resolution image with noninvasion [20]. Originally, UBM was introduced to image the anterior segment of the eye by Canadian professors Pavlin and Foster in the early 1990s [21]. In 1994, a UBM device comprising a 50 MHz probe (a central unit), power adapter, eyecup, and footswitch was produced by the German Zeiss-Humphrey company. UBM features a noninvasive imaging modality that uses ultrasound echoes to produce images. Unlike B-mode ultrasound devices, UBM could improve resolution and detect small structures within biological tissues by using an ultra-high-frequency probe operating at 50 MHz. Currently, UBM has mainly been used for diagnosing human ophthalmic diseases [22,23]. Ultra-high-frequency probes contribute to accurate diagnosis, therapy, and surgery, even in the presence of anatomical or pathological obstacles [24]. 

In vivo microscopy is a highly valuable technique for studying the normal physiological activities of various organs in live animals [25]. The implantation of artificial intravital windows and integration of corresponding microscopic imaging techniques are needed to capture high-resolution images of the target tissue. This technique has been indicated as a strength tool to visualize multiple organs, including the brain [26,27], spinal cord [28], pancreas [29,30], liver [31], lungs [32], colon [33], and ovaries [34]. Furthermore, intravital windows could be maintained in animals’ bodies for a period of time and combined with various biological techniques, which contribute to long-term, stable imaging of organs and tissues [35]. Therefore, this imaging technique plays a vital role in life science and provides a novel insight into the understanding of complex physiological processes in vivo. The application of live imaging windows has mainly been limited to mice, while UBM has primarily been used for diagnosing ophthalmic diseases in humans. Nevertheless, novel insights were obtained by employing these two techniques to visualize visceral organs and tissues in rabbits, including the ovaries [36,37]. However, few studies have been conducted to investigate the development of follicles in vivo by combining live imaging windows with UBM. In the present study, an abdominal intravital window combined with UBM was used to continuously image the ovarian structure in female rabbits. We hope this innovative method could help in providing new insight into ovarian development instead of traditional histological examinations. 

## 2. Materials and Methods

### 2.1. Animals and Equipment

Ten New Zealand White female rabbits (5-month-old, 3.27 ± 0.06 kg, multiparous) were housed in controlled environmental conditions (20 °C, 14:10 h light: dark cycle) and were allowed ad libitum access to food and water. The type of UBM is MD-320W, from MEDA Co., Ltd., Tianjin, China. The medium used is pure water.

### 2.2. Design of Intravital Window

The implantable intravital window is primarily made of polytetrafluoroethylene material. UBM is a B-mode ultrasonic diagnostic equipment, and the presence of a medium is essential for UBM to image. In the present study, the entire component was designed as an openable assembly comprising the main body and cover (Figure 1a) for facilitating image capture of ovaries. The whole diameter of the component is 45 mm, the overall height is 0.4 mm, the outer diameter of the imaging window is 37 mm, and the inner diameter of the imaging window is 35 mm (Figure 1c,d). Of note, the size of the imaging window should be large enough for the operation of the UBM probe. The inner wall of the main body is a threaded connection with a cover, and twelve 2 mm small holes for suturing to the skin and muscles are distributed on the outer side of the main body. The diameter and height of the cover are 45 mm and 7 mm, respectively. The inner diameter of the cover is 32 mm, connecting with the main body. The outer diameter of the cover is larger than the suture hole of the main body which could protect skin incision and prevent rabbits from licking the wound. The intravital window was made by a factory specialized in part processing based on our designed drawings.

### 2.3. Surgical Procedure

As depicted in Figure 2, a meticulous surgical protocol was performed to realize an implantable optical imaging window intended to visualize the ovarian structure. A small animal anesthesia machine and isoflurane were employed to anesthetize the female rabbits. Subsequently, intubation and mechanical ventilation were employed during surgery to ensure survival. The left abdominal region beneath the last rib was shaved, and the surgical site was thoroughly disinfected using an alcohol and iodine solution. Then, an incision through the skin and muscle layers was performed to expose the ovary using a surgical knife to make a 25 mm incision below the last rib on the left side of the abdomen. The incision is 30 mm away from the last rib and orthogonal to it. The ovarian tissue was carefully removed from the abdominal cavity using forceps, and absorbable sutures were used to suture the muscle layer to secure the ovaries outside the muscle layer. Be careful to avoid cutting off blood vessels and other tissues connected to the ovaries to ensure the normal physiological function of the ovaries after surgery.

Furthermore, nonabsorbable sutures were employed to suture the imaging window onto the skin and muscle layers to ensure a tight seal and maintain sterility. Once the accurate placement and fixation of the imaging window were confirmed, the cover was placed. Additionally, wound clearance, monitoring of healing and infection, and overall recovery situation were performed. All the animal experimental procedures were approved by the Institutional Animal Care and Use Committee of China Agricultural University (AW52304202-1-1). Overall, this surgical protocol provides a novel and effective method to implant an optical imaging window for the visualization of the ovarian structure of small animals.

### 2.4. Statistical Analysis

All data were presented as means ± SEMs and were subjected to a one-way analysis of variance (ANOVA; SAS Version 9.1). All statistical charts were generated by GraphPad Prism (Version 10.1.2). Differences between groups were determined using Tukey’s multiple comparisons test. Differences at *p* < 0.05 were considered significant. 

## 3. Results

### 3.1. The Impact of Surgery on Female Rabbit Physiology

To evaluate the process of surgical ovarian translocation and window implantation on the physiology of the rabbit, we randomly selected 5 out of 10 rabbits with blood-collection-determined red blood cell (RBC) and white blood cell (WBC) counts using standard laboratory procedures. Results illustrated in Figure 3a indicate that RBC counts markedly reduced after surgery. Subsequently, RBC counts gradually increased and eventually returned to preoperative levels on the 7th and 9th day (Figure 3a). Similarly, the tendency of WBC counts was similar to that of RBC, indicating a similar response in hematological characteristics (Figure 3a,b).

Additionally, feed intake and body weight change were determined during the experimental period (Figure 3c,d). As shown in Figure 3c, a significant reduction (49.1%) in feed intake was observed on the first day of the experiment (*p* < 0.01) compared with preoperation (−1 day). Starting from the 6th day after surgery, the feed intake was restored to preoperative feed intake (*p* > 0.05). Correspondingly, the body weight changes displayed a similar trend. Furthermore, unusual behavior was not observed during the experimental period. Last but not least, around the 4th day after surgery, there was a gradual accumulation of fascia and adipose tissue at the surgical site, but we can clean it up at any time through the intravital window. Collectively, these results indicate that window implantation surgery exerts transient physiological effects on female rabbits. 

### 3.2. High-Resolution Image of Follicles

An ovarian imaging examination of rabbits was performed after window implantation surgery (Figure 4). In the present study, the resolution of the UBM used could reach 30 μm, and the test results reveal the presence of well-defined anechoic spherical structures (dark areas). The well-defined anechoic spherical structures were identified as antral follicles with a diameter ranging from 0.35 mm to 2.01 mm. Of note, we observed cumulus–oocyte complexes (COCs) in each larger follicle (>0.7 mm). Although it is difficult to observe COCs in smaller follicles, it still provides an opportunity to use UBM in vivo to monitor changes in follicles over time. As shown in Figure 4, the position of each follicle was easily located, and the size of the follicles was measured by UBM software. Captured images were analyzed and processed to locate and track each follicle with the same color circle. The number of detectable follicles in the left ovary of three rabbits is presented in Table 1. Among these detectable follicles, the minimum follicle with a diameter of 0.35 mm and the maximum follicle with a diameter of 2.01 mm were observed (n = 3).

### 3.3. Monitoring of Follicular Development in Female Rabbits

To avoid the stress response caused by blood collection, five rabbits without blood collection were selected to perform ovarian development monitoring (Figure 5). One week after surgery, ovaries were scanned by a UBM device for 6 days in the morning. Daily monitoring revealed fluctuations in the number of antral follicles per rabbit over the observation period (Figure 5a). Notably, variabilities were observed among individual rabbits, with some rabbits showing a consistent increase in antral follicle counts and others exhibiting fluctuating or relatively stable counts. Additionally, the diameter of antral follicles was measured and displayed in Figure 5b. 

## 4. Discussion 

UBM has been used as a valuable tool to observe mouse ovarian structure in vivo [38]. However, UBM cannot be used to image the ovaries of the rabbit in vivo because of the thicker body wall and insufficient penetration of ultra-high-frequency ultrasound. A previous study demonstrated that rabbit ovaries with surgical translocation to a subcutaneous position could be effectively imaged using UBM. However, the subcutaneous fascia and adipose tissue growth in female rabbits after surgery often hinders UBM imaging. Therefore, a second surgery is usually required to remove obstructed tissue [37]. Furthermore, the thick skin of the female rabbit weakened the ultrasound of UBM and reduced the resolution of the image. Based on these points, detailed improvements were conducted in the present study. Briefly, ovaries were removed to the subcutaneous position, and an openable imaging window was implanted onto the ovaries. These two processes facilitate UBM to detect ovarian tissue in vivo. At the same time, this protocol also provides operability for the postoperative removal of some redundant fascia and adipose tissue.

In the present study, the feasibility of utilizing UBM to examine the ovarian structure in live rabbits by surgical ovarian translocation and window implantation was documented. Postoperatively, WBC, RBC, food intake, and weight decreased, and they all returned to the preoperative average level within one week. Although some rabbits showed signs of postoperative infection, it did not lead to severe complications and did not affect the maintenance of ovarian function. These results collectively demonstrate the success of the window implantation surgery.

Daily monitoring of follicle development was performed from the 7th day post operation and lasted for 6 days. Storing images and then recording data is very challenging because it is easy to confuse each follicle position and difficult to distinguish different follicles. Therefore, data were examined immediately. UBM comes with a playback function, and we can record the number and size of follicles through temporarily stored images after scanning once. Variabilities of follicle counts per rabbit were observed. Results indicate that some rabbits showed a consistent increase in follicle counts and others exhibited relatively stable counts. This variability highlights the dynamic nature of follicular development in response to physiological and environmental factors, consistent with normal follicular development in rabbits [39,40]. The follicle images revealed that a combination of UBM with an intravital window is an effective method for real-time monitoring of ovarian follicles, which could provide continuous and noninvasive visualization. Furthermore, long-term monitoring further reveals the ability of this method to assess ovarian structure and function over an extended period. The real-time imaging capabilities of UBM and the accessibility provided by the intravital window provide a novel insight into follicular dynamics in vivo, which may enhance our understanding of ovarian physiology in rabbits. 

Despite the successful implantation of the intravital window, some challenges remain. Previous studies suggest that relocating the ovaries subcutaneously does not affect normal ovarian function within two months [37]. However, infection risk, potential compression of ovarian blood vessels by overextended muscle healing, and other factors influencing ovarian function should be taken into consideration. Therefore, the intravital window should not be retained for an extended period. Based on our results and previous studies, the maintenance of an intravital window in a body is 2–3 weeks, comprehensively considering the surgical environment, disinfection procedures, postoperative care, and so on. Future studies are needed to optimize surgical techniques and postoperative management, prolong the survival of the intravital window, and minimize the risk of adverse effects.

Before conducting this experiment, we hoped to detect the dynamic process of follicle changes. However, in actual monitoring, we found that it is difficult to locate the position of every follicle because of many limitations. Firstly, the UBM probe used in the present study is like a B-ultrasound probe, and manual control is required. Accurate manual force, movement speed, and scanning angle control are necessary during the process. Minor changes in these aspects result in different images during scanning. Secondly, the ovarian follicles of female rabbits are constantly changing every day, and the number of follicles is generally more than 10. Follicular changes are difficult to locate and speculate on based on the scanning images captured in two days. Finally, the duration of each scan of the rabbit is another issue worth paying attention to. UBM monitoring was performed on live rabbits with surgery, and scanning time should be strictly controlled to reduce the stimulation of the rabbits. Therefore, it is difficult for us to adjust the angle and intensity of ovarian scanning using UBM based on the monitoring images from the previous day in a short period, and we can only make a rough adjustment. In this study, the limitations of human operation are the main issue. If UBM had a fixed scanning probe or scanning area that could be mechanically controlled like other imaging devices, the limitations of human operation could be solved. In conclusion, we can only quickly record the size and number of antral follicles. The inability to dynamically present the daily changes in follicles is also a regret in our research. At the same time, this is also the driving force for our future study, which constantly improves experimental methods.

## 5. Conclusions

In the present study, results demonstrate that the combination of UBM and intravital window implantation offers a high-resolution imaging solution to visualize ovarian structure in vivo. The utilization of UBM for continuous ovarian imaging through an intravital window has been proven to be rapid and precise. This advancement plays a key role in replacing labor-intensive and time-consuming conventional methods for monitoring ovarian dynamics. Furthermore, the safety of transferring ovaries to a subcutaneous location was confirmed, which enables the study of dynamic processes for ovaries and reduces the number of rabbits. We anticipate that this combined method will offer distinct advantages in the investigation of rabbit ovarian dynamics, ovulation induction, and COCs.

## Figures and Tables

**Figure 1 animals-14-01727-f001:**
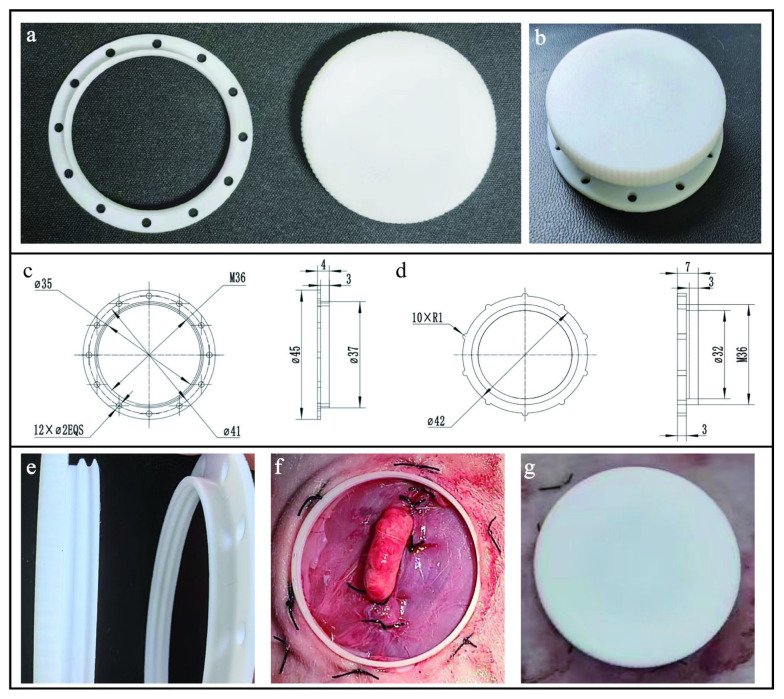
The structure design and usage of the intravital window: (**a**) Images of every part of the intravital window. (**b**) The combination image of the intravital window main body and cover. (**c**) Computer-aided design of main body of the intravital window. All dimensions are expressed in millimeters. (**d**) Computer-aided design of the intravital window cover with an antislip design on the outermost side. All dimensions are expressed in millimeters. (**e**) The inner wall of the window is designed with internal threads connecting to the cover. (**f**,**g**) The intravital window was implanted onto the rabbit ovaries.

**Figure 2 animals-14-01727-f002:**
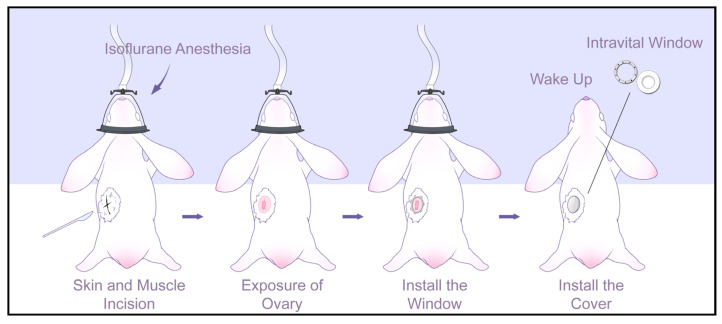
Overview of the surgical protocol of window implantation. The overall surgical procedure required approximately 30 min to conclude.

**Figure 3 animals-14-01727-f003:**
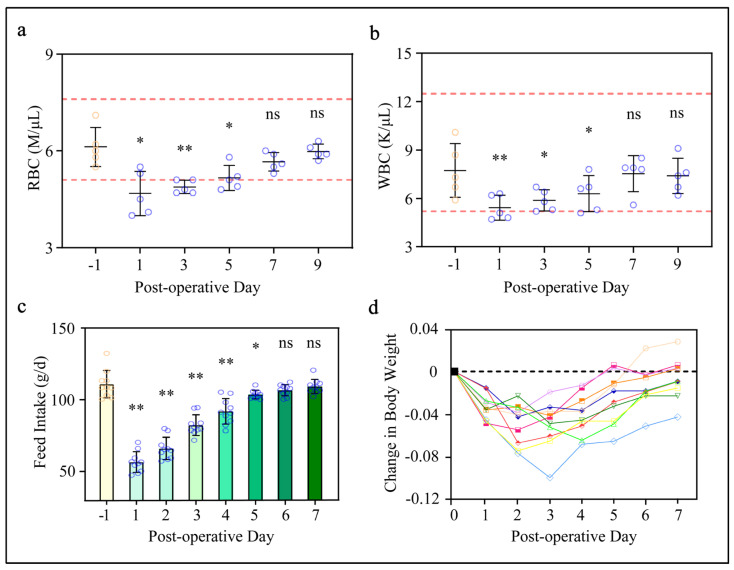
The impact of window implantation surgery on female rabbit physiology: (**a**,**b**) RBC and WBC counts on female rabbits (n = 5) during the experimental period monitored. The red dashed lines represent the normal range of values; (**c**) feed intake from 1 day before surgery (day −1) to the 7th day after surgery (day 7) in rabbits (n = 10) compared with day −1 (** *p* < 0.01,* *p* < 0.05, ns *p* > 0.05); (**d**) relative weight change after surgery (day 0) in rabbits (n = 10) compared with day 0 (** *p* < 0.01,* *p* < 0.05, ns *p* > 0.05).

**Figure 4 animals-14-01727-f004:**
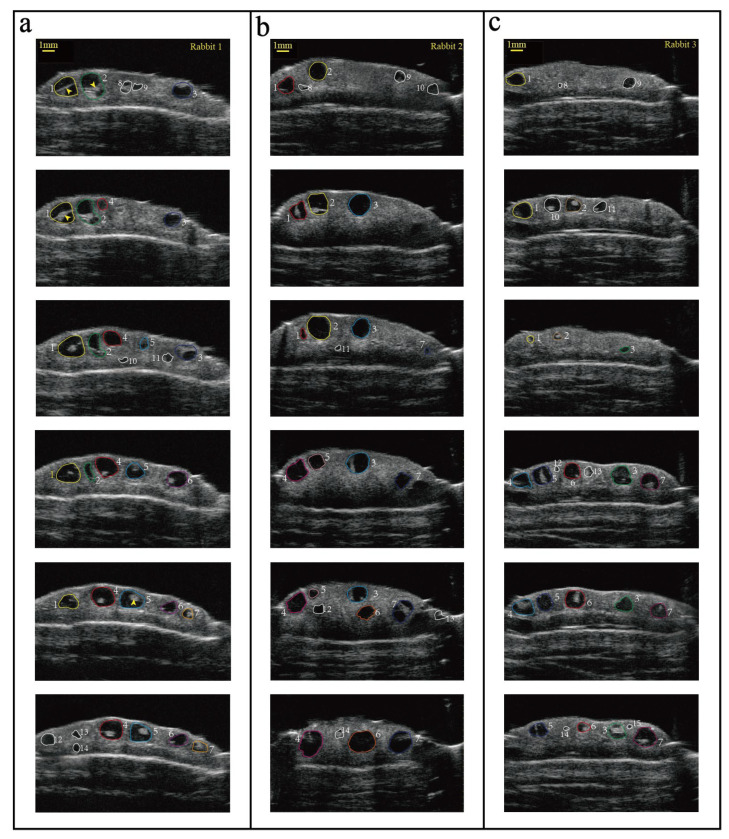
Ovarian follicle imaging captured in vivo by UBM after surgery in three rabbits (**a**–**c**). During the scanning procedure, the rabbits were anesthetized. From one side to the other, imaging of different slices of unilateral exposed ovaries is shown. The same color represents the same follicle in one rabbit, while the white color represents follicles that only appear once in the image. The yellow arrows refer to the cumulus–oocyte complexes in (**a**). Scale bar = 1 mm.

**Figure 5 animals-14-01727-f005:**
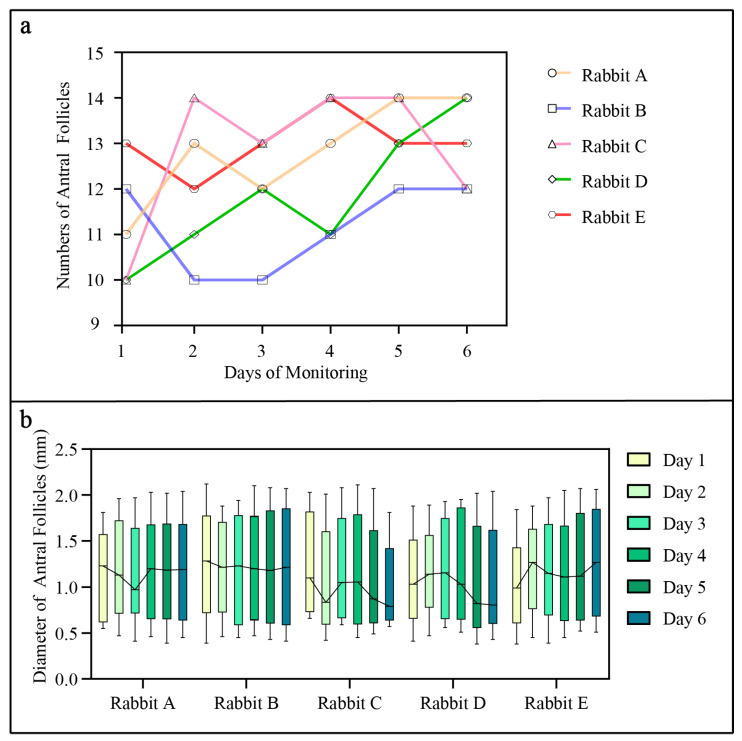
Numbers (**a**) and diameter (**b**) of ovarian antral follicles in female rabbits (n = 5) during six continuous days of examination. Monitoring of antral follicular by UBM was performed without anesthesia starting 7 days after surgical window implantation. In panel (**b**), the box plots indicate the daily distribution of antral follicle diameter in rabbits (n = 5); the continuous lines show the daily changes in follicular mean diameters.

**Table 1 animals-14-01727-t001:** Number and size of antral follicles detected by UBM after surgery in three rabbits (Figure 4a–c).

Rabbit	Ovarian Antral Follicle	
	Numbers	Mean ± SEM (Range)
a	14	1.18 ± 0.14 (0.55–2.01)
b	14	1.19 ± 0.15 (0.43–1.94)
c	15	1.09 ± 0.13 (0.35–1.80)

Note: Minimum and maximum values are in parentheses.

## Data Availability

Data are contained within the article.

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
