# Peer review of "In Vivo Imaging of Rabbit Follicles through Combining Ultrasound Bio-Microscopy and Intravital Window"

_animals, 2024, doi:10.3390/ani14121727_

Round 1
Reviewer 1 Report
Comments and Suggestions for Authors
The Authors report a surgical technique for the implantation of an intra-vital window into the abdominal flank of female rabbits that enable the exteriorization of ovaries and high-resolution imaging of their structures by ultrasound-bio-microscopy (UBM). The Authors examined the ovarian dynamic processes regarding follicles counts and size for six continuous days. According to the Authors, the results “provide a relative new method to monitor ovarian dynamic processes and to understand the reproductive physiology of female rabbits”.
I decided to review this paper based on the title and its aim “... to monitor the ovarian dynamic processes.“ that intrigued me. At the end, I come up somewhat frustrated: I will try to explain you why below under the key concern section. A part this, I found several other issues (listed below under the major concerns) regarding the methodology, results, and discussion as well as several other minor issues.
Key concerns
To me, Figures 5a and 5b do not clearly represent the daily dynamic processes regarding the count and size of antral follicles that occur in the rabbit ovary.
In fact, on a given day following the first observation (Day 1), the total number (n) of (antral) follicles (TF) is equal to the number of pre-existing follicles (PF – found the day before) plus the newly formed follicles (NF) minus regressing follicles (RF), both of which occurred during the observation interval:
nTF = nPF + nNF – nRF (1)
Therefore, taking rabbit C as an example and using (1) on day 2, you could have
14TF = 10PF + 4NF (2)
However, it cannot be excluded that the same result could be obtained (hypothetically) in the following way:
14TF = 10PF + 8NF – 4RF (3)
or any other arithmetic combination of the above variables whose sum gives 14TF.
However, this problem could have been solved because, for each Monitoring Day, it is possible to exactly track the position of each antral follicle using its spatial coordinates (x, y, z) and discriminate between developing and regressing follicles.
Unfortunately, this possibility was not explored by the authors.
Major concerns
Introduction
Lines 35-36: While I agree that rabbits can be used as farm animal (for meat and fur production), experimental animal models as well as pets (three-purpose animals), I remain somewhat disappointed by the cited reference (n.1; Kim, J.M. et al 2022) that is very specific dealing with a case report of pulmonary aneurism. Indeed, I would suggest to provide more appropriated references chosen among the several excellent reviews on these topics.
Lines 36-37. Please, add reference(s), preferably review(s). This is a central point to understand your ovarian follicular data.
Lines 38-40. Some references are very dated: The references 2 and 4, for example, were published in 1930 and 1936. Others are very specific such as that one pinpointing to the role of nerve growth factor (NGF) to document follicle development. Again, while I may agree that the cytokine NGF might be involved in the ovarian physiology, for sure there are several other common hormones (FSH, prolactin, GnRH, ….) that sustain follicular development, estrous behavior, ovulation (in a word: reproductive physiology). Also in these cases, I would suggest to utilize reviews among the many papers published on this filed.
Materials and Methods
Lines 82-84. Specify, at least, the age and weight of rabbits and whether they were nulliparous (virgin) or not (primiparous or multiparous). At the time of in vivo examination with UBM what was the estrous condition of each rabbit? This reproductive clue strictly connects with number and size of ovarian follicles.
Lines 86-on (Design of intravital window). I do not understand whether the implantable intra-vital windows were purchased or homemade using a 3-D printer. Similarly provide the type of the UBM used the type of medium. In any case, please provide all the direct or indirect information needed to repeat your experimental design.
Lines 109-110. Fig. 2 does not show how “to fabricate an implantable optical imaging window …” but rather the steps of the surgical protocol for the ovary translocation and intra vital window implantation.
Line 108-on (Surgical procedure). In general, I found the overall surgical section quite scanty of useful information for anyone intends to use the same experimental approach.
Line 113-114. Looking at Figure 2. It looks like the incision was done in the left, high abdominal flank. How long the incision was? It was parallel or orthogonal in respect to the last rib? In relocating the ovary, all vascular, nervous, and lymphatic connections remained intact?
Line 121. What does it mean “… and overall recovery situation were performed”. I suppose RBC and WBC counts, feed intake and body weight were examined, but how blood samples were obtained? No mention on that.
Line 130 (Statistical analysis). Did you test the normal distribution of your data? Were “all data” continuous or also discrete? How did you perform imaging capture by UBM? Data were saved and stored for later analysis or immediately examined. One or more individuals examined the number of follicles and their position independently from each other.
Results
Line 137-139. Why using RBC and WBC counts to evaluate rabbit physiology after surgery? For physiology do you simply mean normal health condition, or post-surgery-recovery?
Line 145-147. You wrote: “As shown in Figure 3c, a significant reduction (49.1%) in feed intake was observed on the first day of the experiment (P < 0.01) compared with preoperation (-1 day).” Looking at figures 3a-3c I understand that data were collected one day BEFORE surgery and one day AFTER surgery (post-operative Day) and at different intervals thereafter. Therefore, the day of surgery was defined as Day 0 and no blood sample was collected here. I infer this possibility by checking the scale under figure 3d: The weight change after surgery in rabbits (n=10); Compared with -1 (**P < 0.01,*P < 0.05, ns P > 0.05). The body weight change were not compared with -1 day but with day 0. No statistic here? Anyway, I suggest uniforming the plots of figure 3 for a better consistency.
Line 164-167. It looks like you consider all follicles with an anechoic structure as “antral”. Did you observe COCs in all follicles?
Figure 4. I am not sure how to read the six frames sequence for each rabbit: from left to right in the upper and then bottom series or from top and bottom pictures of the left, middle, and right panels. I suggest arranging the panels of each rabbit vertically with a top to down sequence that reconstructs the anatomy of the ovary. What was the thickness of the ovarian tissue at which each was taken?
Table 1. The table can be simplified by specifying that size of follicles are mean +/- SEM (Minimum-Maximum) values (mm). For example
Rabbit 1 14 1.18 ± 0.14 (0.55 – 2.01)
Correct “Title 1” in the left heading of the table
Lines 182-186 and Figure 5a. Please clarify here that you are referring to antral follicles.
Figure 5b. What does it represents this figure? A box plot? Please clarify. Add statistical analysis to data of figure 5a and 5b.
Overall, in “Results” I count 3 rabbits used in the first 7 post-surgery days plus 5 rabbits for daily monitoring of ovarian follicles. Two rabbits are missing. Were they used only for studying feed intake and body weight following surgery? Please clarify.
Discussion
In this section, there are several reiterations of data already shown in results.
At the end, I agree that in the rabbit ovary there is always a pool of follicles ready to ovulate following an appropriate stimulus, either copulation or injection of GnRH or other ovulation inducing factors, but the underlying dynamic processes still remain unresolved despite the effort of the Authors.
Minor concerns
Lines 62-66. You should add also the ovary to the list here provided.
Lines 54-55. It looks like the verb is missing (“UBM features with … by using … to produce images”).
Lines 72-74. “….. to visualize visceral organs and tissues in rabbits (32,33)” please add “including the ovary” which is exactly the target organ of paper 33.
Line 86. Remove “primarily” because it implies that there are also secondary and/or other materials.
Line 109. Use “realize” instead of “fabricate”.
Line 138. Change the period with a comma (… of the rabbit, red blood cells….).
Line 140. I found the term “population” not appropriate. Use, simply, RBC count(s). The same, for WBC.
Line 189. “…. for severely obstructed female rabbits”. Please clarify.
Line 210-216. These sentences repeat information already provided and as such are unnecessary. Comment differently.
Comments on the Quality of English LanguageSome comments included in the previous section
Author Response
Manuscript title:
In Vivo Imaging of Rabbit Follicles through Combining Ultrasound Bio-microscopy and Intravital Window(Manuscript ID: animals-2999005)
The Authors report a surgical technique for the implantation of an intra-vital window into the abdominal flank of female rabbits that enable the exteriorization of ovaries and high-resolution imaging of their structures by ultrasound-bio-microscopy (UBM). The Authors examined the ovarian dynamic processes regarding follicles counts and size for six continuous days. According to the Authors, the results “provide a relative new method to monitor ovarian dynamic processes and to understand the reproductive physiology of female rabbits”.
I decided to review this paper based on the title and its aim “... to monitor the ovarian dynamic processes” that intrigued me. At the end, I come up somewhat frustrated: I will try to explain you why below under the key concern section. A part this, I found several other issues (listed below under the major concerns) regarding the methodology, results, and discussion as well as several other minor issues.
Reply:
Thanks for your constructive comments on our manuscript entitled “In Vivo Imaging of Rabbit Follicles through Combining Ultrasound Bio-microscopy (Manuscript ID: animals-2999005)”. We have carefully considered all the suggestions and concerns raised during the review process, which are quite helpful for us to improve the quality of our manuscript. Our point-by-point responses are shown below and highlighted (red) in revised version.
Key concerns:
To me, Figures 5a and 5b do not clearly represent the daily dynamic processes regarding the count and size of antral follicles that occur in the rabbit ovary.
In fact, on a given day following the first observation (Day 1), the total number (n) of (antral) follicles (TF) is equal to the number of pre-existing follicles (PF – found the day before) plus the newly formed follicles (NF) minus regressing follicles (RF), both of which occurred during the observation interval:
nTF = nPF + nNF – nRF (1)
Therefore, taking rabbit C as an example and using (1) on day 2, you could have
14TF = 10PF + 4NF (2)
However, it cannot be excluded that the same result could be obtained (hypothetically) in the following way:
14TF = 10PF + 8NF – 4RF (3)
or any other arithmetic combination of the above variables whose sum gives 14TF.
However, this problem could have been solved because, for each Monitoring Day, it is possible to exactly track the position of each antral follicle using its spatial coordinates (x, y, z) and discriminate between developing and regressing follicles.
Unfortunately, this possibility was not explored by the authors.
Reply:
Thanks for your detailed concerns and evaluation of our research. We completely agree with your point of view. Before conducting this experiment, we also wanted to detect the dynamic process of follicle changes. However, in actual monitoring, we found that it is difficult to locate position of follicles because of many limitations.
Firstly, UBM probe used in the presents study is similar to a B-ultrasound probe and manual control are required. Accurate manual force, movement speed, and scanning angle control are necessary during using process. Changes in manual force, movement speed, and scanning angle control result in different images during scanning. Secondly, the ovarian follicles of female rabbits are constantly changing every day, and the number of follicles is generally more than 10. Follicular changes are difficult to locate and speculate based on scan images captured in two days. Finally, the duration of each scan of rabbit is another issue worth paying attention. UBM monitoring were performed on live rabbits with surgery and scanning time should be strictly controlled to reduce stimulation to rabbits. Therefore, it is difficult for us to adjust the angle and intensity of ovarian scanning using UBM based on the monitoring images from the previous day in a short period, and we can only make a rough adjustment. Limitations of human operation is the main issue. If UBM owns a fixed scanning probe or scanning area that can be mechanically controlled like other imaging devices, limitations of human operation can be solved.
In conclusion, we can only quickly record the size and the number of antral follicles. The inability to dynamically present the daily changes of follicles is also a regret in our research. At the same time, this is also the driving force for our future study, which constantly improves experimental methods. Finally, thank you again for your suggestions in this regard.
Major concerns:
Introduction
- Lines 35-36: While I agree that rabbits can be used as farm animal (for meat and fur production), experimental animal models as well as pets (three-purpose animals), I remain somewhat disappointed by the cited reference (n.1; Kim, J.M. et al 2022) that is very specific dealing with a case report of pulmonary aneurism. Indeed, I would suggest to provide more appropriated references chosen among the several excellent reviews on these topics.
Reply:
Thanks for your suggestion! We have revised the sentence and cited new literatures:
[1] doi:10.1016/j.meatsci.2018.06.037, [2] doi:10.1016/j.pharmthera.2014.09.009, [3] doi:10.1016/j.theriogenology.2006.04.015), please see details in line 36 and reference list.
- Lines 36-37. Please, add reference(s), preferably review(s). This is a central point to understand your ovarian follicular data.
Reply:
Thanks! The suggested changes have been made: [4] doi:10.1016/j.anireprosci.2011.11.007. Please see details in line 38 and reference list.
- Lines 38-40. Some references are very dated: The references 2 and 4, for example, were published in 1930 and 1936. Others are very specific such as that one pinpointing to the role of nerve growth factor (NGF) to document follicle development. Again, while I may agree that the cytokine NGF might be involved in the ovarian physiology, for sure there are several other common hormones (FSH, prolactin, GnRH, ….) that sustain follicular development, estrous behavior, ovulation (in a word: reproductive physiology). Also in these cases, I would suggest to utilize reviews among the many papers published on this filed.
Reply:
Thanks! We have made modifications to the literature citation based on your suggestion.
We have replaced outdated literature 3 and 5. Then, we have added new reference: [5] doi:10.1016/j.anireprosci.2010.10.003, [11] doi:10.1016/j.anireprosci.2014.06.001) in the field of endocrinology. Please see details in line 39, 40 and reference list.
Materials and Methods
- Lines 82-84. Specify, at least, the age and weight of rabbits and whether they were nulliparous (virgin) or not (primiparous or multiparous). At the time of in vivo examination with UBM what was the estrous condition of each rabbit? This reproductive clue strictly connects with number and size of ovarian follicles.
Reply:
Thanks! We have added some information according to your suggestion. Please see details in line 83.
Three female rabbits used in this section were mainly between the proestrus and estrus periods. The UBM imaging results of female rabbits at different stages of the estrus cycle are presented in other experimental results.
- Lines 86-on (Design of intravital window). I do not understand whether the implantable intra-vital windows were purchased or homemade using a 3-D printer. Similarly provide the type of the UBM used the type of medium. In any case, please provide all the direct or indirect information needed to repeat your experimental design.
Reply:
Thanks! The intravital window was made by a factory specialized in part processing based on our designed drawings. The UBM model is MD-320W, from MEDA Co., Ltd. Tianjin, China. The medium used is pure water. We added these descriptions in line 85 and 86.
- Lines 109-110. Fig. 2 does not show how “to fabricate an implantable optical imaging window …” but rather the steps of the surgical protocol for the ovary translocation and intra vital window implantation.
Reply:
Thank you for pointing out the mistake in our manuscript. We added “Using a surgical knife to make a 25 mm incision below the last rib on the left side of the abdomen. The incision is 30 mm away from the last rib and orthogonal to it.” to make description appropriate. Please see details in line 117-119.
- Line 108-on (Surgical procedure). In general, I found the overall surgical section quite scanty of useful information for anyone intends to use the same experimental approach.
Reply:
Thank you for your suggestion. Because this surgery is relatively simple, the steps were written relatively succinctly. Of course, we modified the surgical section based on your suggestion. Please see details in line117-123.
- Line 113-114. Looking at Figure 2. It looks like the incision was done in the left, high abdominal flank. How long the incision was? It was parallel or orthogonal in respect to the last rib? In relocating the ovary, all vascular, nervous, and lymphatic connections remained intact?
Reply:
Thanks for your comments. The incision is on the left abdomen of the rabbit, 30 mm away from the last rib. The incision is 25 mm and orthogonal to the last rib. We did not cut off the tissue connected to the ovary during ovarian translocation in order to ensure the normal physiological function of the ovary. Therefore, vascular, nervous, and lymphatic connections remained intact.
- Line 121. What does it mean “… and overall recovery situation were performed”. I suppose RBC and WBC counts, feed intake and body weight were examined, but how blood samples were obtained? No mention on that.
Reply:
Thanks! "Overall recovery situation was performed" here refers to "RBC and WBC counts, feed intake, and body weight". We have added blood collection description in line143-145.
- Line 130 (Statistical analysis). Did you test the normal distribution of your data? Were “all data” continuous or also discrete? How did you perform imaging capture by UBM? Data were saved and stored for later analysis or immediately examined. One or more individuals examined the number of follicles and their position independently from each other.
Reply:
Thanks for your suggestions. All data follows a normal distribution and is continuous. Data were examined immediately, because UBM comes with a playback function. After scanning once, we can record the number and size of follicles through temporarily stored images. Storing data and then recording data is very challenging because it is easy to confuse each follicle position and difficult to distinguish different follicles.
Results
- Line 137-139. Why using RBC and WBC counts to evaluate rabbit physiology after surgery? For physiology do you simply mean normal health condition, or post-surgery-recovery?
Reply:
Thanks for your comments. We referred to a method described in a previous study (doi: 10.1038/nmeth.4511.). They used WBC and RBC as indexes to evaluate postoperative physiology. Physiology here refers to the postoperative recovery situation.
- Line 145-147. You wrote: “As shown in Figure 3c, a significant reduction (49.1%) in feed intake was observed on the first day of the experiment (P < 0.01) compared with preoperation (-1 day).” Looking at figures 3a-3c I understand that data were collected one day BEFORE surgery and one day AFTER surgery (post-operative Day) and at different intervals thereafter. Therefore, the day of surgery was defined as Day 0 and no blood sample was collected here. I infer this possibility by checking the scale under figure 3d: The weight change after surgery in rabbits (n=10); Compared with -1 (**P < 0.01,*P < 0.05, ns P > 0.05). The body weight change were not compared with -1 day but with day 0. No statistic here? Anyway, I suggest uniforming the plots of figure 3 for a better consistency.
Reply:
Thanks!We provided a detailed explanation for your question.
Blood collection performed one day before surgery (-1 day), with an interval of one day. No blood was collected on the day of the surgery.
The feed intake on the day of surgery was not recorded because each rabbit started the surgery at a different time, so their feeding time on the day of surgery is different, which can affect the statistical results.
We performed weighing from the day of surgery before anesthesia, so it is marked as 0 day. Although we strongly agree with your point that it is best to maintain consistency in the figure, we believe that recording the postoperative weight changes from the day of surgery before the anesthesia is more reflective.
Thank you very much for your suggestions.
- Line 164-167. It looks like you consider all follicles with an anechoic structure as “antral”. Did you observe COCs in all follicles?
Reply:
Thanks! Similar to conventional ultrasound, UBM mainly forms images based on the different absorption levels of ultrasound in various tissues and the return signals. The follicular cavity in the ovary is filled with fluid, so the absorption of ultrasound is the weakest, resulting in a silencing structure. COCs has a higher density, so it is easy to distinguish. We observed COCs in every larger follicle (>0.7mm), but it is difficult to observe COCs in smaller follicles.
- Figure 4. I am not sure how to read the six frames sequence for each rabbit: from left to right in the upper and then bottom series or from top and bottom pictures of the left, middle, and right panels. I suggest arranging the panels of each rabbit vertically with a top to down sequence that reconstructs the anatomy of the ovary. What was the thickness of the ovarian tissue at which each was taken?
Reply:
Thanks! Based on your suggestion, we changed the arrangement order of the images in Figure 4. The thickness of each ovarian tissue is less than 5 mm. Please see details in Figure 4.
- Table 1. The table can be simplified by specifying that size of follicles are mean +/- SEM (Minimum-Maximum) values (mm). For example
Rabbit 1 14 1.18 ± 0.14 (0.55 – 2.01)
Reply:
Thanks! We changed the table 1 according to your suggestions. Your valuable idea makes table 1 more precise and clearer. Please see details in Table 1.
- Correct “Title 1” in the left heading of the table
Reply:
Thank you for pointing out our mistake! We changed it. Please see details in Table 1.
- Lines 182-186 and Figure 5a. Please clarify here that you are referring to antral follicles.
Reply:
Thanks! We changed description based on your suggestions. Please see details in line 191, 193, 194 and Figure 5a.
- Figure 5b. What does it represent this figure? A box plot? Please clarify. Add statistical analysis to data of figure 5a and 5b.
Reply:
Thanks!This is a box plot of the size of observable antral follicles in 6 days of monitoring from 5 rabbits (Rabbit A, Rabbit B, Rabbit C, Rabbit D, Rabbit E). We used GraphPad Prism to calculate and generate line chart and box plot in figure 5a and 5b.
- Overall, in “Results” I count 3 rabbits used in the first 7 post-surgery days plus 5 rabbits for daily monitoring of ovarian follicles. Two rabbits are missing. Were they used only for studying feed intake and body weight following surgery? Please clarify.
Reply:
Thanks!We provided detailed explanation for your suggestion.
We performed surgery with 10 rabbits. Three rabbits among 10 rabbits were randomly selected to perform with detailed scans under anesthesia after surgery. These results were presented in Table 1 and Figure 4 (results section 3.2). After surgery, feed intake and weight of all rabbits were recorded and five rabbits among 10 rabbits were randomly selected for blood collection. These results were descripted in result 3.1 section. Correspondingly, five rabbits without blood collection were conducted with UBM monitoring in order to eliminate the possibility of stress interference caused by blood collection. Results were illustrated in result section 3.3.
Discussion
In this section, there are several reiterations of data already shown in results.
At the end, I agree that in the rabbit ovary there is always a pool of follicles ready to ovulate following an appropriate stimulus, either copulation or injection of GnRH or other ovulation inducing factors, but the underlying dynamic processes still remain unresolved despite the effort of the Authors.
Reply:
Thanks for your suggestion! We rewritten the duplicated parts in discussion based on your suggestion.
The primary purpose of our manuscript is to propose a new method for studying female rabbit follicles. The study of follicle dynamics was included in our next examination, and some technical limitations still need to be overcome. Therefore, research on follicle dynamics related to ovulation and other aspects has yet to be mentioned in this manuscript.
Minor concerns
- Lines 62-66. You should add also the ovary to the list here provided.
Reply:
Thanks! We have made modifications and added corresponding literature based on your suggestions: [34] doi:10.1016/j.celrep.2021.109382, please see details in line 66.
- Lines 54-55. It looks like the verb is missing (“UBM features with … by using … to produce images”).
Reply:
We are sorry for our mistake. We changed this sentence to “UBM features a non-invasive imaging modality that uses ultrasound echoes to produce images.” Please see details in line 54 and 55.
- Lines 72-74. “….. to visualize visceral organs and tissues in rabbits (32,33)” please add “including the ovary” which is exactly the target organ of paper 33.
Reply:
Thanks! We have added “including the ovary” by your suggestion, please see details in line 74.
- Line 86. Remove “primarily” because it implies that there are also secondary and/or other materials.
Reply:
Thanks! We have already deleted “primarily” by your suggestion. Please see details in line 88.
- Line 109. Use “realize” instead of “fabricate”.
Reply:
Thanks! We have made changes. Please see details in line 110.
- Line 138. Change the period with a comma (… of the rabbit, red blood cells….).
Reply:
Thanks! We have changed this sentence to “To evaluate the process of surgical ovarian translocation and window implantation on the physiology of the rabbit, we selected 5 of 10 rabbits (n = 5) with surgery for blood collection and determined red blood cell (RBC) counts and white blood cell (WBC) count.” by your previous suggestion. Please see details in line 142-144.
- Line 140. I found the term “population” not appropriate. Use, simply, RBC count(s). The same, for WBC.
Reply:
Thanks! We have replaced “population” with RBC count(s) or WBC count(s).
- Line 189. “…. for severely obstructed female rabbits”. Please clarify.
Reply:
We are sorry for the confused description and changed this sentence to “However, the subcutaneous fascia and adipose tissue growth in female rabbits after surgery often hinders UBM imaging. Therefore, a second surgery is usually required to remove obstructed tissue.” Please see details in line 204-206.
- Line 210-216. These sentences repeat information already provided and as such are unnecessary. Comment differently.
Reply:
Thanks! We have made changes. Please see details in line 214-219.

Reviewer 2 Report
Comments and Suggestions for Authors
This manuscript described a combination of UBM and intravital window implantation for high resolution imaging of ovarian structure in vivo, and it offers distinct advantages on the investigation of ovarian dynamics, ovulation induction, and COCs in rabbit. This manuscript was well designed and presented. These are some issues should be addressed.
1. Detailed information such as the manufacturer and model of the UBM needs to be indicated.
2. Please list the detailed steps for UBM detection of rabbit ovaries and follicles.
3. What is the estrus cycle of 10 experimental rabbit? Why only 3 rabbit were used listed in table 1 and figure 4 and 5 rabbit were used in figure 5?
4. It is difficult to determine the dynamic development process of follicles by only monitoring the development of follicles in 3 rabbits for 6 days.
Author Response
Manuscript title:
In Vivo Imaging of Rabbit Follicles through Combining Ultrasound Bio-microscopy and Intravital Window
Manuscript ID: animals-2999005
This manuscript described a combination of UBM and intravital window implantation for high resolution imaging of ovarian structure in vivo, and it offers distinct advantages on the investigation of ovarian dynamics, ovulation induction, and COCs in rabbit. This manuscript was well designed and presented. These are some issues should be addressed.
Reply:
Thanks for your constructive comments on our manuscript entitled “In Vivo Imaging of Rabbit Follicles through Combining Ultrasound Bio-microscopy (Manuscript ID: animals-2999005)”. We carefully considered all the issues raised during the review process. Our point-by-point responses are shown below and highlighted (red) in revised version.
- Detailed information such as the manufacturer and model of the UBM needs to be indicated.
Reply:
Thanks! We added information about the manufacturer and model of the UBM in line 85-86.
- Please list the detailed steps for UBM detection of rabbit ovaries and follicles.
Reply:
Thanks! First, we placed the rabbit on the fixing device and open the window cover. Then, we gently rinse the periphery of the ovaries with physiological saline. Using the UBM probe to slowly move the scan from one side of the ovary to the other. UBM equipped with a playback function. After scanning once, number and size of follicles through temporarily stored images were determined. Data were examined immediately. After the data recording was completed, window was recovered with cover and disinfected with iodophor.
- What is the estrus cycle of 10 experimental rabbit? Why only 3 rabbit were used listed in table 1 and figure 4 and 5 rabbit were used in figure 5?
Reply:
Thanks! In the experiments involved in this manuscript, we did not conduct estrus cycle detection of rabbits.
The data involved in Table 1 are the scan results of randomly selected female rabbits who have just undergone surgery. This is the statistical result of the size of each follicle in the three ovaries that received scanning in Figure 4.
We performed surgery with 10 rabbits. Three rabbits among 10 rabbits were randomly selected to perform with detailed scans under anesthesia after surgery. Theses results were presented in Table 1 and Figure 4 (results section 3.2). After surgery, feed intake and weight of all rabbits were recorded and five rabbits among 10 rabbits were randomly selected for blood collection. These results were descripted in result 3.1 section. Correspondingly, five rabbits without blood collection were conducted with UBM monitoring in order to eliminate the possibility of stress interference caused by blood collection. Results were illustrated in result section 3.3.
- It is difficult to determine the dynamic development process of follicles by only monitoring the development of follicles in 3 rabbits for 6 days.
Reply:
Thanks! We agree with your point.
The primary purpose of our manuscript is to propose a new method for studying female rabbit follicles. The study of follicle dynamics was included in our next examination, and some technical limitations still need to be overcome. Therefore, research on follicle dynamics related to ovulation and other aspects has yet to be mentioned in this manuscript. Only the monitoring results of follicles from 5 rabbits within 6 days are presented in this manuscript.

Round 2
Reviewer 1 Report
Comments and Suggestions for Authors
While the authors have addressed appropriately most of my questions, concerns, and suggestions, after careful re-review of changes made (thanks for your efforts), I have still some recommendations that I think necessary to improve the overall quality of the paper.
I am personally convinced that figures and tables are the first and most read parts of a paper. Consequently, I think that these essential parts should be highly accurate in conveying the desired message.
For example:
Figure 2. Overview of the surgical protocol ….. implantation. The overall surgical procedure required approximately …… to conclude.
Figure 3. The impact of window implantation surgery on female rabbit physiology. RBC (a) and WBC (b) counts on female rabbits (n = 5) during the experimental period monitored; The red dashed lines in panels a and b represent the normal range of values;……. d) Relative weight change after surgery (day 0) in rabbits (n=10); Compared with day 0 (**P 165 < 0.01,*P < 0.05, ns P > 0.05).
Figure 4. Ovarian follicles captured in vivo by UBM X days after surgery in three rabbits (a-c). During the scanning procedure, the rabbits were anesthetized. From top to down, imaging of different slices of unilateral exposed ovaries. Each slice is approximately ??? µm thick. The same color represents the same follicle in one rabbit, while white color represents follicles that only appear once in the image. The yellow arrows refer to the cumulus-oocyte complexes in Figure 4a. Scar bar = 1 mm.
Table 1. Number and size of follicles detected by UBM X days after surgery in three rabbits (a-c). In parenthesis, minimum and maximum values. (Aren't these 3 rabbits the same as those described in figure 4? If yes, as I suppose, please change the table and text accordingly.
Heading:
_____________________________________
Rabbit Ovarian antral follicle
# Number Mean ± SEM (range)
______________________________________
a 14 1.18 ± 0.14 (0.55 – 2.01)
b
c
_____________________________________
Figure 5. Number (panel a) and diameter (panel b) of ovarian antral follicles in female rabbits (n = 5) during six continuous days of examination. Monitoring of antral follicular by UBM was performed without anesthesia starting X days after surgical window implantation. In panel B, the box plots indicate (PLEASE SPECIFY); the continuous lines show the daily changes of follicular mean diameters.
Why you do not report your own comments in the paper where appropriate? Below a short list:
1) We observed COCs in every larger follicle (>0.7mm), but it is difficult to observe COCs in smaller follicles.
2) We used GraphPad Prism to calculate and generate line chart and box plot in figure 5a and 5b (Under the statistical section)
3) Thanks for your comments. We referred to a method described in a previous study (doi: 10.1038/nmeth.4511.). They used WBC and RBC as indexes to evaluate postoperative physiology. Physiology here refers to the postoperative recovery situation. I cannot find this reference when appropriate, that is where you use RBC and WBC counts to assess surgery recovery.
4) Data were examined immediately, because UBM comes with a playback function. After scanning once, we can record the number and size of follicles through temporarily stored images. Storing data and then recording data is very challenging because it is easy to confuse each follicle position and difficult to distinguish different follicles. This comments strike with what reported for rabbits a-c, but it is very important for the reader and should be commented .
5) The intravital window was made by a factory specialized in part processing based on our designed drawings.
6) Before conducting this experiment, we also wanted to detect the dynamic process of follicle changes. However, in actual monitoring, we found that it is difficult to locate position of follicles because of many limitations. Firstly, UBM probe used in the presents study is similar to a B-ultrasound probe and manual control are required. Accurate manual force, movement speed, and scanning angle control are necessary during using process. Changes in manual force, movement speed, and scanning angle control result in different images during scanning. Secondly, the ovarian follicles of female rabbits are constantly changing every day, and the number of follicles is generally more than 10. Follicular changes are difficult to locate and speculate based on scan images captured in two days. Finally, the duration of each scan of rabbit is another issue worth paying attention. UBM monitoring were performed on live rabbits with surgery and scanning time should be strictly controlled to reduce stimulation to rabbits. Therefore, it is difficult for us to adjust the angle and intensity of ovarian scanning using UBM based on the monitoring images from the previous day in a short period, and we can only make a rough adjustment. Limitations of human operation is the main issue. If UBM owns a fixed scanning probe or scanning area that can be mechanically controlled like other imaging devices, limitations of human operation can be solved. In conclusion, we can only quickly record the size and the number of antral follicles. The inability to dynamically present the daily changes of follicles is also a regret in our research. At the same time, this is also the driving force for our future study, which constantly improves experimental methods. (I find these comments very honest and, as such, they should be reported in your paper at the end of discussion under a paragraph dedicated to the limitations of the experimental protocol and future improvements).
Line 144. …. were randomly selected 5 out of 10 rabbits with ……. determined red blood cell (RBC) and white blood cell (WBC) counts using standard laboratory procedures.
Lines 205-207. I do not understand if this is a direct experimental evidence following surgery in 10 rabbits or is a cited sentence. Please clarify in both cases. If you encountered these post-surgical “complications” specify (in Results), at least, in how many cases and after how many days they occurred.
In the abstract correct "population"
Author Response
Manuscript title:
In Vivo Imaging of Rabbit Follicles through Combining Ultrasound Bio-microscopy and Intravital Window
Manuscript ID: animals-2999005
While the authors have addressed appropriately most of my questions, concerns, and suggestions, after careful re-review of changes made (thanks for your efforts), I have still some recommendations that I think necessary to improve the overall quality of the paper.
I am personally convinced that figures and tables are the first and most read parts of a paper. Consequently, I think that these essential parts should be highly accurate in conveying the desired message.
Reply:
Thanks for the constructive comments on our manuscript entitled “In Vivo Imaging of Rabbit Follicles through Combining Ultrasound Bio-microscopy and Intravital Window (Manuscript ID: animals-2999005)”. We have carefully considered all the suggestions raised during the review process, which are quite helpful for us to improve the overall quality of the paper. Our point-by-point responses are shown and highlighted (green) in revised version.
- Figure 2. Overview of the surgical protocol ….. implantation. The overall surgical procedure required approximately …… to conclude.
Reply:
Thanks! We modified this figure legend according to your suggestion.
- Figure 3. The impact of window implantation surgery on female rabbit physiology. RBC (a) and WBC (b) counts on female rabbits (n = 5) during the experimental period monitored; The red dashed lines in panels a and b represent the normal range of values ;……. d) Relative weight change after surgery (day 0) in rabbits (n = 10); Compared with day 0 (**P < 0.01, *P < 0.05, ns P > 0.05).
Reply:
Thanks! We modified this figure legend according to your suggestion.
- Figure 4. Ovarian follicles captured in vivo by UBM X days after surgery in three rabbits (a-c). During the scanning procedure, the rabbits were anesthetized. From top to down, imaging of different slices of unilateral exposed ovaries. Each slice is approximately ??? µm thick. The same color represents the same follicle in one rabbit, while white color represents follicles that only appear once in the image. The yellow arrows refer to the cumulus-oocyte complexes in Figure 4a. Scar bar = 1 mm.
Reply:
Thanks! We modified figure legend of Figure 4 according to your suggestion. We are very sorry for the missing information on thickness of slice. UBM's scanning mode does not display the spacing among images, so thickness of each slice was not provided.
- Table 1. Number and size of follicles detected by UBM X days after surgery in three rabbits (a-c). In parenthesis, minimum and maximum values. (Aren't these 3 rabbits the same as those described in figure 4? If yes, as I suppose, please change the table and text accordingly.
Reply:
Thanks! These 3 rabbits are the same as those described in figure 4. We changed the table and text based on your valuable suggestion. Please see details in Figure 4 and Table 1.
- Figure 5. Number (panel a) and diameter (panel b) of ovarian antral follicles in female rabbits (n = 5) during six continuous days of examination. Monitoring of antral follicular by UBM was performed without anesthesia starting X days after surgical window implantation. In panel B, the box plots indicate (PLEASE SPECIFY); the continuous lines show the daily changes of follicular mean diameters.
Reply:
Thanks! We modified figure legend of Figure 5 according to your suggestion.
Why you do not report your own comments in the paper where appropriate? Below a short list:
Reply:
Thanks for your suggestion! We added our comments in the paper where appropriate. The following point-by-point responses were provided.
1) We observed COCs in every larger follicle (>0.7mm), but it is difficult to observe COCs in smaller follicles.
Reply:
Thanks! We have added comment for this description, please see details in line 173-176.
2) We used GraphPad Prism to calculate and generate line chart and box plot in figure 5a and 5b (Under the statistical section)
Reply:
Thanks! We provided statistical analysis information in line 138-139.
3) Thanks for your comments. We referred to a method described in a previous study (doi: 10.1038/nmeth.4511.). They used WBC and RBC as indexes to evaluate postoperative physiology. Physiology here refers to the postoperative recovery situation. I cannot find this reference when appropriate, that is where you use RBC and WBC counts to assess surgery recovery.
Reply:
Thanks! We referred to a previous study published in Nature Method (Entenberg D, Voiculescu S, Guo P, Borriello L, Wang Y, Karagiannis GS, Jones J, Baccay F, Oktay M, Condeelis J. A permanent window for the murine lung enables high-resolution imaging of cancer metastasis. Nat Methods. 2018 Jan;15(1):73-80) which determined physiological impact of the window for high-resolution imaging of the lung by performing a series of experiments looking at reactivity to handling, appearance, behavior, weight, and counts of white and red blood cells.
4) Data were examined immediately, because UBM comes with a playback function. After scanning once, we can record the number and size of follicles through temporarily stored images. Storing data and then recording data is very challenging because it is easy to confuse each follicle position and difficult to distinguish different follicles. This comments strike with what reported for rabbits a-c, but it is very important for the reader and should be commented.
Reply: Thanks! We have added comments to the Discussion. We have changed the expression of this sentence. Please see details in line 229-233.
5) The intravital window was made by a factory specialized in part processing based on our designed drawings.
Reply:
Thanks! We changed as your suggestion, please see details in line 100 and 101.
6) Before conducting this experiment, we also wanted to detect the dynamic process of follicle changes. However, in actual monitoring, we found that it is difficult to locate position of follicles because of many limitations. Firstly, UBM probe used in the presents study is similar to a B-ultrasound probe and manual control are required. Accurate manual force, movement speed, and scanning angle control are necessary during using process. Changes in manual force, movement speed, and scanning angle control result in different images during scanning. Secondly, the ovarian follicles of female rabbits are constantly changing every day, and the number of follicles is generally more than 10. Follicular changes are difficult to locate and speculate based on scan images captured in two days. Finally, the duration of each scan of rabbit is another issue worth paying attention. UBM monitoring were performed on live rabbits with surgery and scanning time should be strictly controlled to reduce stimulation to rabbits. Therefore, it is difficult for us to adjust the angle and intensity of ovarian scanning using UBM based on the monitoring images from the previous day in a short period, and we can only make a rough adjustment. Limitations of human operation is the main issue. If UBM owns a fixed scanning probe or scanning area that can be mechanically controlled like other imaging devices, limitations of human operation can be solved. In conclusion, we can only quickly record the size and the number of antral follicles. The inability to dynamically present the daily changes of follicles is also a regret in our research. At the same time, this is also the driving force for our future study, which constantly improves experimental methods. (I find these comments very honest and, as such, they should be reported in your paper at the end of discussion under a paragraph dedicated to the limitations of the experimental protocol and future improvements).
Reply: Thanks for your suggestion! We added these comments in Discussion section , please see details in line 254-273.
7) Line 144. …. were randomly selected 5 out of 10 rabbits with ……. determined red blood cell (RBC) and white blood cell (WBC) counts using standard laboratory procedures.
Reply:
Thanks! We have added this comment in line 144-146.
8) Lines 205-207. I do not understand if this is direct experimental evidence following surgery in 10 rabbits or is a cited sentence. Please clarify in both cases. If you encountered these post-surgical “complications” specify (in Results), at least, in how many cases and after how many days they occurred.
Reply:
Thanks! These cases occurred in the cited reference (Cervantes, 2013). We changed the citation position of this reference (line 211 to line 214). These cases also occurred in our experiment and these tissues can be cleaned up at any time in case of the openable window. We added information in Discussion section (line 218-220). Also, we added and explained this case in Results 3.1 (line 157-159).
9) In the abstract correct "population"
Reply:
Thank you for pointing out this error in manuscript. We made revisions in the Abstract section (line 26).
